# Anti-Biofilm Activity of Oleacein and Oleocanthal from Extra-Virgin Olive Oil toward *Pseudomonas aeruginosa*

**DOI:** 10.3390/ijms25095051

**Published:** 2024-05-06

**Authors:** Marisa Di Pietro, Simone Filardo, Roberto Mattioli, Giuseppina Bozzuto, Giammarco Raponi, Luciana Mosca, Rosa Sessa

**Affiliations:** 1Department of Public Health and Infectious Diseases, “Sapienza” University, p.le Aldo Moro, 5, 00185 Rome, Italy; marisa.dipietro@uniroma1.it (M.D.P.); simone.filardo@uniroma1.it (S.F.); giammarco.raponi@uniroma1.it (G.R.); 2Department of Biochemical Sciences, Faculty of Pharmacy and Medicine, “Sapienza” University, p.le Aldo Moro, 5, 00185 Rome, Italy; roberto.mattioli@uniroma1.it (R.M.); luciana.mosca@uniroma1.it (L.M.); 3National Centre for Drug Research and Evaluation, Italian National Institute of Health, 00161 Rome, Italy; giuseppina.bozzuto@iss.it

**Keywords:** oleacein, oleocanthal, *Pseudomonas aeruginosa*, antibiotic resistance, anti-biofilm activity

## Abstract

New antimicrobial molecules effective against *Pseudomonas aeruginosa*, known as an antibiotic-resistant “high-priority pathogen”, are urgently required because of its ability to develop biofilms related to healthcare-acquired infections. In this study, for the first time, the anti-biofilm and anti-virulence activities of a polyphenolic extract of extra-virgin olive oil as well as purified oleocanthal and oleacein, toward *P. aeruginosa* clinical isolates were investigated. The main result of our study was the anti-virulence activity of the mixture of oleacein and oleocanthal toward multidrug-resistant and intermediately resistant strains of *P. aeruginosa* isolated from patients with ventilator-associated pneumonia or surgical site infection. Specifically, the mixture of oleacein (2.5 mM)/oleocanthal (2.5 mM) significantly inhibited biofilm formation, alginate and pyocyanin production, and motility in both *P. aeruginosa* strains (*p* < 0.05); scanning electron microscopy analysis further evidenced its ability to inhibit bacterial cell adhesion as well as the production of the extracellular matrix. In conclusion, our results suggest the potential application of the oleacein/oleocanthal mixture in the management of healthcare-associated *P. aeruginosa* infections, particularly in the era of increasing antimicrobial resistance.

## 1. Introduction

Over the years, the exploration of natural bioactive nutraceuticals as supplementation or alternative therapeutic options has acquired great importance for their good efficacy and better safety profiles than synthetic drugs [1,2,3,4,5].

Olive-tree-based molecules have a clear history of beneficial properties for human health, typically associated with the phenolic contents in leaves, drupes, and extra-virgin olive oil (EVOO) [6]. In particular, EVOO, obtained from the cold pressing of *Olea europaea* L. drupes, a cornerstone of the Mediterranean diet, is widely studied for its nutritional properties and as a source of nutraceuticals, including oleacein and oleocanthal, which are able to modulate the onset and progression of metabolic, cardiovascular, and neurological diseases, as well as cancer [3,7,8]. However, oleacein and oleocanthal isolation from EVOO is cumbersome and expensive. Recently, the use of natural deep eutectic solvents (NaDESs), combined with other techniques, has been implemented to obtain high amounts of oleocanthal and oleacein in a fast, easy, cheap, and eco-friendly manner [9].

EVOO extracts and purified oleacein and oleocanthal have also been suggested as potential strategies to counteract the phenomenon of antibiotic resistance, one of the greatest threats to human health [3,10].

Among the bacteria with growing multidrug resistance, the WHO included ESKAPE pathogens (*Enterococcus faecium*, *Staphylococcus aureus*, *Klebsiella pneumoniae*, *Acinetobacter baumannii*, *Pseudomonas aeruginosa*, and *Enterobacter* species), against which new antibiotics are urgently needed. *Pseudomonas aeruginosa* is one of the most prevalent ESKAPE pathogens in hospital environments, causing more than 50% of the healthcare-acquired infections, including ventilator-associated pneumonia, intensive care unit infections, and surgical site infections, with mortality rates of 20–60% [11,12,13,14,15,16].

*P. aeruginosa* is described to secrete different virulence factors, playing an essential role in pathogenesis as well as in antibiotic resistance [14,17]. One of the main virulence factors of *P. aeruginosa* is the biofilm, a bacterial community embedded in an exopolysaccharide matrix, known to contribute to therapeutic failure and to escape from the immune system, resulting in chronic infections [13,18,19]. Indeed, bacteria grown into biofilms are approximately 10–1000 times more resistant to antibiotics than planktonic cells, and this seems to be related to alginate production in mucoid strains of *P. aeruginosa* [13,18,20,21]. Another important and unique virulence factor of *P. aeruginosa* is pyocyanin, which has been described to contribute to host tissue damage as well as to biofilm formation by promoting bacterial cell-to-cell interactions [22,23]. Also, *P. aeruginosa* motility has been correlated to adaptive resistance to antibiotics and to biofilm formation because it was found that non-motile bacterial mutants could lack the ability to form biofilms [24,25].

Herein, for the first time, the anti-biofilm and anti-virulence activities of a polyphenolic extract of EVOO as well as purified oleocanthal and oleacein, dissolved in NaDES, have been investigated toward *P. aeruginosa* isolated from patients with ventilator-associated pneumonia or surgical site infection.

## 2. Results

### 2.1. Phenotypic Characterization of Clinical P. aeruginosa Strains

Preliminary identification revealed that two clinical strains of *P. aeruginosa*, isolated from bronchoalveolar lavage fluid (PA BL) and from a surgical infection site (PA SW), were mucoid and non-mucoid strains, respectively (Table 1). By analyzing the antibiotic susceptibility patterns, multidrug resistance was observed in the PA BL strain; the PA BL strain was resistant, overall, to nine antibiotics belonging to cephalosporins, beta-lactam inhibitor combinations, fluoroquinolones, aminoglycosides, and carbapenems (Appendix A). By contrast, the PA SW strain showed an intermediate resistance to approximately 50% of all the tested antibiotics, including cephalosporins, beta-lactam inhibitor combinations, fluoroquinolones, and carbapenem (Appendix A).

In addition, both the multidrug-resistant strain (PA BL) and the intermediately resistant isolate (PA SW) of *P. aeruginosa* were able to produce biofilms, alginate, and pyocyanin as well as spread over the agar surface by swarming and swimming motilities (Table 1). Interestingly, significantly higher alginate and pyocyanin productions were observed in the multidrug-resistant PA BL strain as compared to the intermediately resistant PA SW isolate (*p* < 0.05). Further, PA BL showed a higher swarming motility than PA SW (*p* < 0.005); instead, the swimming motility was similar in both strains.

### 2.2. Antibacterial Activities of OOP, Oleacein, and Oleocanthal

Initially, the antibacterial activity of the NaDES (betaine/propylene glycol) alone was examined against multidrug-resistant (PA BL) and intermediately resistant (PA SW) strains of *P. aeruginosa*, at concentrations ranging from 20% to 2.5% (*v*/*v*). No antibacterial activity of the NaDES toward the *P. aeruginosa* strains was observed at or below a concentration of 20% (*v*/*v*). Then, the antibacterial activities of the OOP as well as the oleacein and oleocanthal were investigated. OOP showed antibacterial activities toward multidrug-resistant (PA BL) and intermediately resistant (PA SW) strains with MIC and MBC values of 3390 µg/mL. Also, oleacein and oleocanthal in the NaDES were effective against PS BL and PS SW strains (MIC/MBC 10 mM). Lastly, the time-kill assay, performed on both *P. aeruginosa* strains, showed that OOP, oleacein, and oleocanthal, at MIC/2 and MIC/4, did not cause a reduction in the bacterial vitality of ≥3 log up to 24 h. Differently, all three compounds were able to reduce the bacterial vitality by ≥3 log at MIC concentrations at 6 h for OOP and 12 h for oleacein and oleocanthal (Appendix A).

### 2.3. Anti-Biofilm Activities of OOP, Oleacein, and Oleocanthal

First, the NaDES alone did not show any anti-biofilm activity against multidrug-resistant (PA BL) and intermediately resistant (PA SW) strains of *P. aeruginosa* at the above concentrations (from 20% to 2.5% *v*/*v*).

Based on the MIC determination results, OOP was used at 1695 µg/mL (MIC/2) to inhibit biofilm formation (Figure 1). OOP treatment of multidrug-resistant (PA BL) and intermediately resistant (PA SW) strains, significantly inhibited biofilm formation when compared with untreated cultures (PA BL *p* = 0.0000013; PA SW *p* = 0.000029) at similar inhibition rates (PA BL 53.2% and PA SW 51.4%). However, the OOP extract had no effect on mature biofilms (Figure 1).

Purified oleacein and oleocanthal dissolved in the NaDES were used at concentrations of 5 mM (MIC/2) and 2.5 mM (MIC/4) to inhibit biofilm formation. As shown in Figure 2, oleacein (5 mM) inhibited the biofilm formation of both strains of *P. aeruginosa* as compared to untreated cells (PA BL *p* = 0.00003; PA SW *p* = 0.048); a stronger inhibitory effect was observed against PA BL (71.4%) as compared to PA SW (33%), although it did not reach statistical significance (Figure 2A). Also, the oleocanthal treatment (5 mM) of multidrug-resistant (PA BL) and intermediately resistant (PA SW) strains of *P. aeruginosa* significantly inhibited biofilm formation when compared with untreated cultures (PA BL *p* = 0.000000009; PA SW *p* = 0.000000001); a higher inhibitory activity of biofilm formation was observed toward the PA BL strain (84.8%) as compared to PA SW (70.7%), although it did not reach statistical significance. The mixture of oleacein/oleocanthal in the NaDES was able to significantly inhibit the biofilm formations of multidrug-resistant (PA BL) and intermediately resistant (PA SW) strains of *P. aeruginosa* when compared with untreated cultures, up to a concentration of 2.5 mM (PA BL: oleacein (5 mM)/oleocanthal (5 mM), *p* = 0.000000004; oleacein (2.5 mM)/oleocanthal (2.5 mM), *p* = 0.000000002; PA SW: oleacein (5 mM)/oleocanthal (5 mM), *p* = 0.0000000006; oleacein (2.5 mM)/oleocanthal (2.5 mM), *p* = 0.00002); a higher inhibitory effect of the mixture of oleacein (2.5 mM)/oleocanthal (2.5 mM) was observed against PA SW (89.7%) as compared to PA BL (77.2%), although it did not reach statistical significance (Figure 2B).

SEM analysis further confirmed the ability of the mixture of oleocanthal (2.5 mM)/oleacein (2.5 mM) in the NaDES to inhibit the biofilm formation of the multidrug-resistant strain (PA BL) and intermediately resistant isolate (PA SW) of *P. aeruginosa*. Indeed, the mixed treatment inhibited bacterial cell adhesion in both strains of *P. aeruginosa*, hindering the production of the extracellular matrix; in PA SW, the remaining fragments of the extracellular matrix also showed a different morphology as compared to untreated cells. On the contrary, untreated biofilms of the multidrug-resistant strain (PA BL) and intermediately resistant isolate (PA SW) of *P. aeruginosa* displayed as aggregates composed of bacterial cells and the extracellular matrix, which was denser and more compact in the PA BL strain as compared to the PA SW isolate (Figure 3).

As for the anti-mature biofilm effects, oleocanthal, at a concentration of 5 mM, was only effective against the multidrug-resistant strain (PA BL) as compared to untreated cultures (*p* = 0.00001, percentage of inhibition: 70.1%), whereas oleacein, at a concentration of 5 mM, did not show any effect on either strain (Figure 4A). The mixture of oleacein (5 mM)/oleocanthal (5 mM) showed an activity similar to that of oleocanthal against mature biofilms (PA BL, *p* = 0.000005, percentage of inhibition: 74.4%) (Figure 4B).

### 2.4. Anti-Virulence Activities of OOP, Oleacein, and Oleocanthal

OOP, at a concentration of 1695 µg/mL, was also able to slightly inhibit alginate production by both multidrug-resistant (PA BL) and intermediately resistant (PA SW) strains of *P. aeruginosa* (inhibition rate: ~11%) as compared to untreated cultures (*p* = NS) (Figure 5A). Moreover, OOP (1695 µg/mL) significantly reduced the pyocyanin production of the PA SW and PA BL strains of *P. aeruginosa* as compared to untreated cells (PA SW: *p =* 0.01; PA LW: *p* = 0.008); a higher inhibition in the production of pyocyanin was observed against the multidrug-resistant (PA BL) strain (56%) as compared to the intermediately resistant (PA SW) strain of *P. aeruginosa* (30%), although it did not reach statistical significance (Figure 5B). Lastly, the OOP extract (169.5 µg/mL) caused strong reductions in swimming and swarming motilities of both strains in comparison to untreated cultures (PA BL swimming, *p* = 0.04, swarming, *p* = 0.002; PA SW swimming, *p* = 0.03, swarming, *p* = 0.014). In particular, the OOP extract reduced the swimming zones of the multidrug-resistant (PA BL) and intermediately resistant (PA SW) strains of *P. aeruginosa* by 80.1% and 91.3%, respectively, whereas lower reductions were observed in the swarming zones (PA BL: 59.8%; PA SW: 55.5%) (Figure 5C).

Oleacein (2.5 mM) significantly inhibited only the alginate production of PA SW as compared to untreated cultures (*p* = 0.02), reducing it by 77%. Oleocanthal, at a concentration of 2.5 mM, was able to significantly inhibit the alginate productions of the multidrug-resistant (PA BL) and intermediately resistant (PA SW) strains of *P. aeruginosa* as compared to untreated cultures (*p* = 0.02), reducing them by 77% and 72%, respectively (Figure 6A).

Interestingly, the mixed oleacein (2.5 mM)/oleocanthal (2.5 mM) treatments of the PA BL and PA SW strains of *P. aeruginosa* significantly inhibited alginate production as compared to untreated cultures (PA SW: *p* = 0.02; PA BL: *p* = 0.03), with similar reduction percentages (PS BL: 62%; PS SW: 64%) (Figure 6A).

As regards other virulence factors, oleacein (2.5 mM) was only able to inhibit pyocyanin production in the PA SW strain as compared to untreated cultures (*p* = 0.009), reducing it by 30%. Oleocanthal (2.5 mM) was able to inhibit pyocyanin production by the multidrug-resistant (PA BL) and intermediately resistant (PA SW) strains of *P. aeruginosa* as compared to untreated cultures (PA BL: *p* = 0.008; PA SW: *p* = 0.004), with the highest effect toward PS BL (inhibition percentages: PA BL, 57%; PA SW, 39%, *p* = NS). Lastly, the mixture of oleacein (2.5 mM)/oleocanthal (2.5 mM) strongly inhibited pyocyanin production in the PA SW and PA BL strains as compared to untreated cultures (PA BL: *p* = 0.01; PA SW: *p* = 0.006), with percentages of inhibition of 55.2 and 39.6, respectively (*p* = NS) (Figure 6B).

Concerning the effects on the motility, in PA BL, oleacein, oleocanthal, and their mixture (0.5 mM) were capable to significantly reduce both the swimming and the swarming zones up to 76% (Figure 6C; *p* < 0.05). By contrast, in PA SW, only oleacein and the mixture of oleacein/oleocanthal (0.5 mM) showed efficacy toward swimming and swarming motilities, with a reduction in the motility zone of up to 80% (Figure 6C; *p* < 0.05).

## 3. Discussion

To date, targeting bacterial virulence factors, such as biofilms, has been a promising approach in the fight against antimicrobial resistance. Indeed, it is estimated that biofilms are responsible for 70% of all the microbial infections [26] and represent one of the major global challenges to control healthcare-acquired infections because of their resistance to antimicrobials and host defense mechanisms [27,28,29].

The main result of our study is the anti-virulence activity of a mixture of two polyphenols isolated from EVOO, namely, oleacein and oleocanthal, toward multidrug-resistant and intermediately resistant strains of *P. aeruginosa*, at sub-MIC concentrations (<10 mM). In particular, the mixture of oleacein (2.5 mM)/oleocanthal (2.5 mM) was the only formulation that significantly inhibited all the virulence factors, including the biofilm formations, alginate and pyocyanin productions, and motilities, in both *P. aeruginosa* strains at up to 90%, 63%, 40%, and 80%, respectively. The inhibitory activity of the oleacein/oleocanthal mixture toward the biofilms was confirmed by SEM analysis; the treatment inhibited bacterial cell adhesion and the production of the extracellular matrix in both strains of *P. aeruginosa*, further supporting its ability to hinder the early steps of biofilm formation. Differently, oleacein and oleocanthal alone inhibited the biofilm formations of both strains of *P. aeruginosa*, only when used at a higher concentration (5 mM). Interestingly, both oleacein and oleocanthal showed stronger inhibitory effects against PA BL (71.4% and 84.8%, respectively) than PA SW (33% and 70.7%, respectively), although they did not reach statistical significance. The low activities of these compounds against PA SW may be linked to differences among the bacterial strains in their genetic background and in the production of other polysaccharides (polysaccharide synthesis loci) and molecules (pyoverdine and rhamnolipids), as well as in the presence of extracellular DNA in the biofilm matrix. All these factors are mainly involved in the initial attachment of bacterial cells and early biofilm formation [30,31].

The discovery of the mixture of oleacein/oleocanthal as a novel anti-biofilm and anti-virulence agent toward *P. aeruginosa* strains isolated from a patient with ventilator-associated pneumonia and from a surgical site infection is of great clinical importance. First, biofilms are known to grow on endotracheal tubes in mechanically ventilated patients or sutures in surgically treated patients and act as bacterial reservoirs, contributing to reinfection and chronic inflammation and, hence, to tissue damage and resistance to treatment [32]. Second, *P. aeruginosa* accounts for 10–20% of isolates in cases of ventilator-associated pneumonia, with an estimated mortality of 32–42.8% [33]. Lastly, surgical site infections are the third most frequent healthcare-associated infections and remain significant causes of morbidity, prolonged hospitalization, and death, with increases from 2- to 11-fold in the risk of mortality [34].

The anti-biofilm and anti-virulence activities of the mixture of oleacein/oleocanthal are even more relevant in light of the fact that in recent years, *P. aeruginosa* has been listed as an antibiotic-resistant “high-priority pathogen” by the WHO [16]. In our study, a mucoid respiratory strain of *P. aeruginosa* was resistant toward several antibiotic classes, including aminoglycosides, and this may be, in part, related to its ability to produce biofilms, which are known to act as physical barriers to antibiotics. In the literature, there is evidence that negatively charged polysaccharides, such as the alginate of *P. aeruginosa*, can effectively sequestrate positively charged aminoglycosides, thus preventing them from penetrating the deeper layers of the biofilm [35].

A further interesting finding of our study is the ability of the oleocanthal (5 mM) to disaggregate the 24 h mature biofilm of the multidrug-resistant respiratory strain but not the biofilm produced by the surgical infection site’s isolate; such activity may be due to differences in the chemical compositions of the biofilms of both strains, as evidenced by alginate assays. Indeed, the respiratory strain was able to produce more alginate than the surgical infection site’s isolate (*p* = 0.048). Consequently, it is likely that the extracellular matrix of the biofilm of the strain isolated from the surgical infection site can include not only alginate but also other polysaccharides, such as pellicle polysaccharide, that are essential for late-stage biofilm formation and maturation [31,35,36,37] and may be less susceptible to oleocanthal.

In our study, the reduced pyocyanin production and bacterial motilities (swarming and swimming), both related to the pathogenicity of *P. aeruginosa*, further support the potential application of the mixture of oleacein/oleocanthal as an anti-biofilm agent. Indeed, both pyocyanin production and bacterial motilities have been described to have significant impacts on biofilm formation; pyocyanin interacts with extracellular DNA, a key component in *P. aeruginosa* biofilms, with the formation of a complex that can stabilize its structure by mediating aggregation and cell–cell interactions within biofilms [38]. Lastly, the initial phases of the *P. aeruginosa* biofilm’s formation are related to bacterial swimming and swarming motilities because the movements of single cells and the coordinated movements of bacteria allow for their adhesion to surfaces [39].

Overall, biofilm formations, alginate and pyocyanin productions, as well as bacterial motilities may represent promising targets by which the oleacein/oleocanthal mixture may reduce bacterial colonization on biotic and abiotic surfaces to counteract the phenomenon of the antibiotic resistance of *P. aeruginosa*. In this scenario, treatment with the oleacein/oleocanthal mixture, at sub-MIC concentrations, may also favor the host’s immune system eliminating less-virulent strains of *P. aeruginosa* and, at the same time, may limit the development of resistance because our polyphenols do not target the pathogen’s growth.

In conclusion, our results indicated that the mixture of oleacein and oleocanthal inhibited the formation of biofilms by downregulating alginate synthesis, pyocyanin production, and swarming and swimming motilities, suggesting its potential application in the management of healthcare-associated *P. aeruginosa* infections, particularly in the era of increasing antimicrobial resistance. Further studies will be necessary to understand the cellular and molecular mechanisms through which these compounds exert their anti-biofilm effects.

## 4. Materials and Methods

### 4.1. Antimicrobial Agents

Olive oil polyphenols (OOPs) were prepared from EVOO, obtained from Coratina cultivar plants, as previously described [10]. Briefly, a natural deep eutectic solvent (NaDES) composed of betaine and propylene glycol mixed in a 1:3.5 molar ratio was used to extract polyphenols from EVOO. A volume of 1 mL of the NaDES was used for each 50 mL of olive oil; the mixture was stirred for 1 h at room temperature and then it was decanted overnight in a separatory funnel. The total polyphenol content of the extract was assayed using the Folin–Ciocalteu reagent (16.95 ± 2.22 mg/mL), whereas its chromatographic profile was obtained by UHPLC-DAD/MS. Oleacein and oleocanthal were found to represent approximately 80% of the total polyphenols [10]. Purified oleacein and oleocanthal, provided by Active-Italia S.r.l., were dissolved in the NaDES at a final concentration of 50 mM (16.0 mg/mL and 15.2 mg/mL for oleacein and oleocanthal, respectively).

### 4.2. Bacterial Strains and Growth Conditions

Two *P. aeruginosa* clinical strains were isolated from the bronchoalveolar lavage fluid (PA BL) of a patient with ventilator-associated pneumonia and from a surgical wound (PA SW) of a patient with surgical site infections. Clinical specimens, investigated at the Microbiology Unit of the “Policlinico Umberto I” Hospital at “Sapienza” University in Rome, were cultured in accordance with guidelines approved by the management of the hospital for routine care purposes. Then, both isolates were identified by matrix-assisted laser desorption ionization–time-of-flight mass spectrometry (MALDI-TOF, Bruker, Bremen, Germany).

Bacterial strains were streaked from stock cultures stored at −80 °C onto tryptic soy agar (TSA) and incubated, for 24 h, at 37 °C.

### 4.3. Phenotypic Characterization of Clinical P. aeruginosa Strains

#### 4.3.1. Antibiotic Susceptibility Assay

Antimicrobial susceptibility testing was performed using a VITEK 2 System (bioMérieux Inc., Marcy-l’Étoile, France) and antimicrobial panels provided by the manufacturer for Gram-negative bacteria (Vitek 2 AST-N397; Microscan NMDRM1). The results were interpreted according to the EUCAST clinical breakpoints [40]. Multidrug resistance was defined, in accordance with the European Centre for Disease Prevention and Control, as a lack of susceptibility to at least one agent in three or more antibiotic categories [41].

#### 4.3.2. Biofilm Assay

*P. aeruginosa* overnight cultures in brain–heart infusion broth (BHI, Oxoid, Basingstoke, UK) were diluted to 1 × 10^8^ CFU)/mL in fresh BHI using a photometric device (optical density at 620 nm). Then, bacterial suspensions were further diluted in BHI to yield a bacterial cell density of 10^5^ CFU/mL.

In 96-well flat-bottom microtiter plates, 50 µL of the bacterial suspension was added to wells prefilled with BHI, and negative vehicle controls (BHI) were also included. After static 24 h and 48 h incubations at 37 °C, the media were aspirated gently, and the wells were washed three times with sterile double-distilled water and thoroughly dried. Then, each well was stained with 100 µL of 0.1% crystal violet and incubated for 15 min at room temperature, rinsed twice with double-distilled water, and thoroughly dried. The adherent biofilm was solubilized with 20% (*v*/*v*) glacial acetic acid and 80% (*v*/*v*) ethanol. After 30 min of incubation at room temperature, the total biofilm biomass in each well was spectrophotometrically quantified at 590 nm [42].

#### 4.3.3. Alginate Assay

The alginate production was determined by incubating *P. aeruginosa* cultures (10^5^ CFU/mL), prepared as above described, at 37 °C for 24 h. Then, the bacteria were removed by centrifugation (10,000 rpm, 15 min), and the supernatant was used for alginate quantification in accordance with the method described by Zheng et al. [43]. By mixing 50 µL of CaCl_2_ (60 mM) and 500 µL of the supernatant, a hydrogel-like alginate aggregate was obtained. Subsequently, the alginate aggregate was soaked in 200 µL of 0.3% crystal violet (CV) dye aqueous solution for 20 min. The CV dye solution was discarded, and the alginate aggregate was rinsed three times with deionized water and dissolved with acetic acid. Then, the alginate production was spectrophotometrically quantified at 600 nm. In every set of experiments, a negative vehicle control (BHI) was also included.

#### 4.3.4. Pyocyanin Assay

*P. aeruginosa* cultures (10^5^ CFU/mL) were prepared as described above and incubated at 37 °C for 24 h. Then, the bacteria were removed by centrifugation (10,000 rpm, 15 min), and pyocyanin was determined as described by Pejec et al. [44]. Briefly, the pyocyanin was extracted by mixing the supernatant with chloroform (1:2 ratio) for 2 min by inversion. Then, the lower layer containing the pyocyanin was transferred to a tube containing 0.2 M HCl in a ratio of 1:2 and mixed for 2 min by inversion. The pink-colored upper layer was separated, and pyocyanin was spectrophotometrically quantified at 520 nm.

#### 4.3.5. Motility Assay

The swimming and swarming motilities of the *P. aeruginosa* clinical strains were evaluated. Briefly, 2 µL of each overnight bacterial suspension, prepared as above described, was inoculated in the center of the agar (swimming agar (1.0% peptone, 0.5% sodium chloride, and 0.3% Bacto-Agar)), and the plates were incubated at 37 °C, according to the protocol described by [45]. Then, after 24 h, the swimming and swarming zones were measured in square centimeters.

### 4.4. Antibacterial Activities of OOP, Oleacein, and Oleocanthal

The antibacterial activities of the OOP, oleacein, and oleocanthal were investigated via antimicrobial susceptibility and time-kill assays.

Antimicrobial susceptibility testing: The minimum inhibitory concentrations (MICs) of the OOP and of the pure compounds were determined using microdilution assays in Muller–Hinton (MH) broth, as previously described [46]. Serial twofold dilutions of the OOP (total polyphenol content of the stock solution: 16,950 μg/mL) were prepared using MHB with polyphenol concentrations ranging from 6780 μg/mL to 424 μg/mL. The concentrations of oleacein and oleocanthal in the NaDES were twofold serial dilutions ranging from 10 mM to 1.25 mM. At the same time, a dilution series of the NaDES (betaine/propylene glycol) was assayed to exclude its potential inhibitory effects. The MIC value was determined as the lowest concentration of the OPP or pure compounds able to inhibit the visible growth of each bacterial strain after 24 h of incubation at 37 °C [47]. To determine the minimum bactericidal concentration (MBC), the lowest concentration capable for inhibiting bacterial growth on an agar surface was evaluated [46]. In every set of experiments, a positive control (gentamicin, 16 μg/mL), negative vehicle control (antimicrobial agent and MHB), and culture control (MHB and bacteria only) were included.

Time-kill assay: A time-kill assay was performed according to CLSI recommendations (https://clsi.org/standards/products/microbiology/documents/m26/, accessed on 16 April 2024). Briefly, a standardized inoculum (1–2 × 10^6^ CFU/mL), prepared from overnight growth on MHB, was exposed to several concentrations (MIC, MIC/2, and MIC/4) of each compound prepared in MHB. Control samples were prepared similarly without exposure to antibacterial agents. At prefixed times of incubation at 37 °C (1, 2, 4, 6, 12, and 24 h), aliquots were removed, and 10-fold serial dilutions were prepared in phosphate buffered saline (PBS) for colony counting. The results were expressed by plotting Log (CFU/mL) over time. The bactericidal activity was defined as a ≥ 3 log (CFU/mL) reduction.

### 4.5. Anti-Biofilm Activities of OOP, Oleacein, and Oleocanthal

#### 4.5.1. Biofilm Inhibition Assay

*P. aeruginosa* overnight cultures in brain–heart infusion broth (BHI, Oxoid, Basingstoke, UK) were diluted to 1 × 10^8^ CFU)/mL in fresh BHI using a photometric device (optical density at 620 nm). Then, bacterial suspensions were further diluted in BHI to yield a bacterial cell density of 10^5^ CFU/mL.

In 96-well flat-bottom microtiter plates, 50 µL of the bacterial suspension was added to wells prefilled with BHI in the presence or in the absence of antimicrobial agents (50 µL). Negative vehicle controls (antimicrobial agent and BHI) were also included. After static 24 h incubation at 37 °C, the biofilm formation was evaluated as above described.

#### 4.5.2. Mature Biofilm Eradication Assay

In 96-well flat-bottom microtiter plates, 50 µL of the bacterial suspension, prepared as above described, was added to wells prefilled with BHI (50 µL). After static 24 h incubation at 37 °C, non-adherent bacteria were removed by washing three times with sterile double-distilled water. Then, adherent cells were incubated in the presence and in the absence of antimicrobial agents (100 µL) for an additional 24 h (48 h in total) at 37 °C. Finally, the total biofilm biomass in each well was determined as described above.

### 4.6. Anti-Virulence Activities of OOP, Oleacein, and Oleocanthal

#### 4.6.1. Alginate Assay

The alginate production was determined by incubating *P. aeruginosa* cultures (10^5^ CFU/mL) in the presence of antimicrobial agents at 37 °C for 24 h, as above described. In every set of experiments, a negative vehicle control (antimicrobial agent and BHI) and a culture control (BHI and bacteria only) were included.

#### 4.6.2. Pyocyanin Assay

*P. aeruginosa* cultures (10^5^ CFU/mL) were incubated in the presence of antimicrobial agents at 37 °C for 24 h and then pyocyanin was determined as above described. In every set of experiments, a negative vehicle control (antimicrobial agent and BHI) and a culture control (BHI and bacteria only) were included.

#### 4.6.3. Motility Assay

The swimming and swarming motilities of the *P. aeruginosa* clinical strains were evaluated in the presence of antimicrobial agents. Briefly, the antimicrobial agents, diluted in BHI, were added to molten swimming and swarming agar, followed by the inoculation of 2 µL of the overnight bacterial suspension at the center of the plates, as above described. In every set of experiments, a negative vehicle control (antimicrobial agent and BHI) and a culture control (BHI and bacteria only) were included.

### 4.7. Scanning Electron Microscopy

In 24-well flat-bottom microtiter plates, 250 µL of the bacterial suspension, prepared as above described, was placed on 12 mm diameter glass coverslips and incubated at 37 °C in the presence or absence of antimicrobial agents (250 µL). After static 24 h incubation, the media were aspirated gently, and coverslips were washed three times with sterile double-distilled water and thoroughly dried. Then, the biofilm-coated coverslips were fixed in 2.5% glutaraldehyde in 0.1 M cacodylate buffer (pH 7.4) at room temperature (RT) for 30 min, post-fixed with 1% OsO_4_ in the same buffer for 1 h, and dehydrated through a graded ethanol series. Ethanol was gradually substituted by a 1:1 solution of hexamethyldisilazane (HMDS)/absolute ethanol for 30 min and successively by pure HMDS for 1 h at RT. Samples were completely dried by removing the HMDS and leaving at RT for 2 h. Dried samples were mounted on stubs, gold-coated by sputtering (SCD040 Balzers device, Bal-Tec, Los Angeles, CA, USA), and analyzed with a field emission gun scanning electron microscope (SEM-FEG, Quanta 200 Inspect, FEI Company, Eindhoven, The Netherlands).

### 4.8. Statistical Analysis

The results were expressed as means ± standard deviation (SD) of three replicates from three independent experiments. Two-way ANOVA was performed for the analysis of variance. All the statistical calculations were performed in Microsoft Excel software (version 2110), using the Real Statistics Resource Pack (release 7.9.1, https://www.real-statistics.com, accessed on 15 March 2022). A value of *p* ≤ 0.05 was considered as being statistically significant.

## Figures and Tables

**Figure 1 ijms-25-05051-f001:**
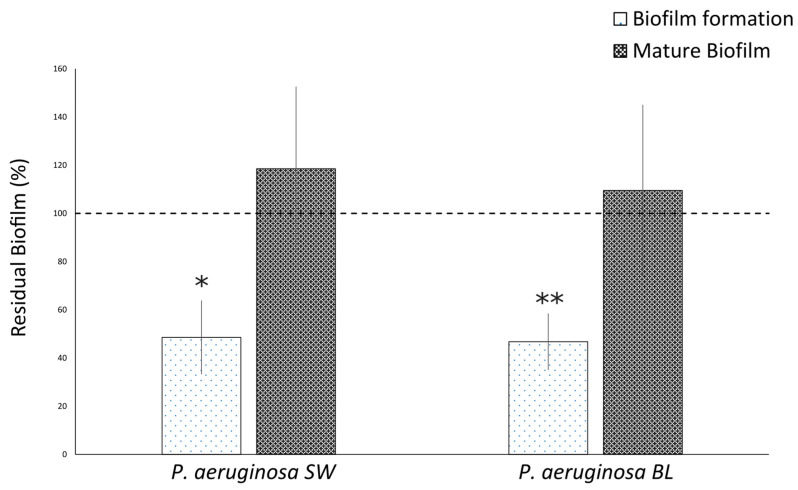
Anti-biofilm activity of OOP in NaDES (1695 μg/mL) toward *P. aeruginosa* isolates: effects on biofilm formation and on mature biofilm. SW, surgical wound (intermediate drug resistance); BL, bronchoalveolar lavage fluid (multidrug resistance). Data are expressed as percentages of residual biofilms ± standard deviation as compared to untreated cells. Dotted line represents the percentage of residual biofilm in untreated *P. aeruginosa*. *, *p* < 0.001 and **, *p* < 0.0001 vs. untreated cells.

**Figure 2 ijms-25-05051-f002:**
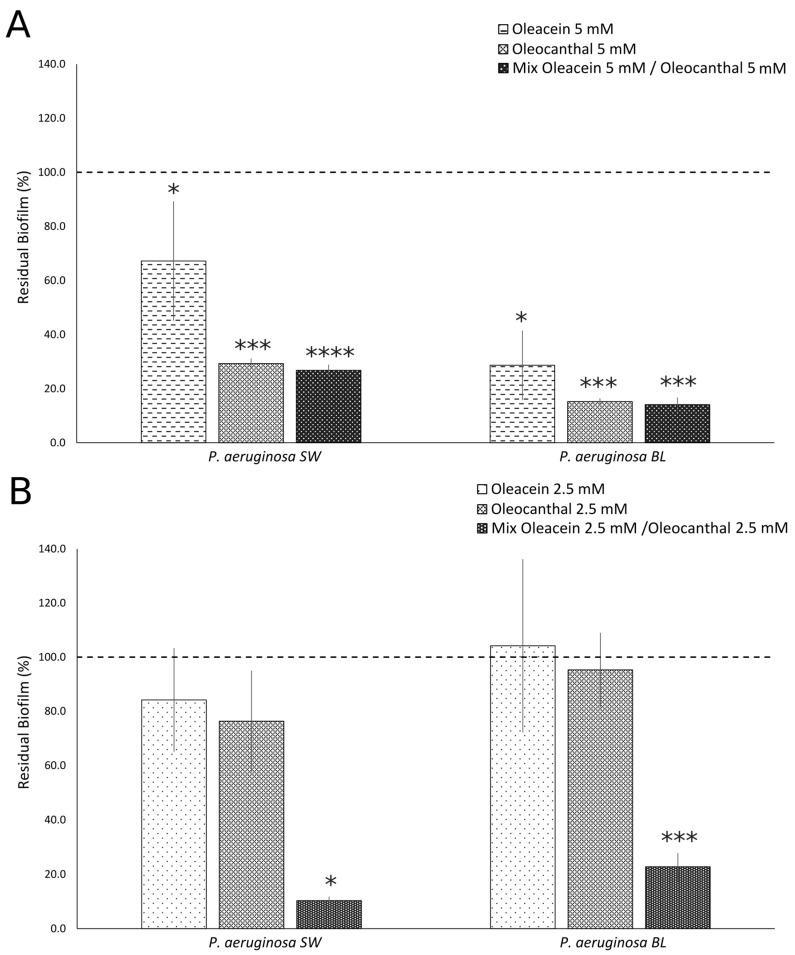
Effects of purified oleacein and oleocanthal in NaDES toward biofilm formation of *P. aeruginosa* isolates. Oleacein, oleocanthal, and their mixture were tested at 5 mM (**A**) and 2.5 mM (**B**) concentrations. SW, surgical wound (intermediate drug resistance); BL, bronchoalveolar lavage fluid (multidrug resistance). Data are expressed as percentages of residual biofilms ± standard deviation as compared to untreated cells. Dotted lines represent the percentage of residual biofilm in untreated *P. aeruginosa*. *, *p* < 0.001; ***, *p* < 0.00000001; ****, *p* < 0.000000001 vs. untreated cells.

**Figure 3 ijms-25-05051-f003:**
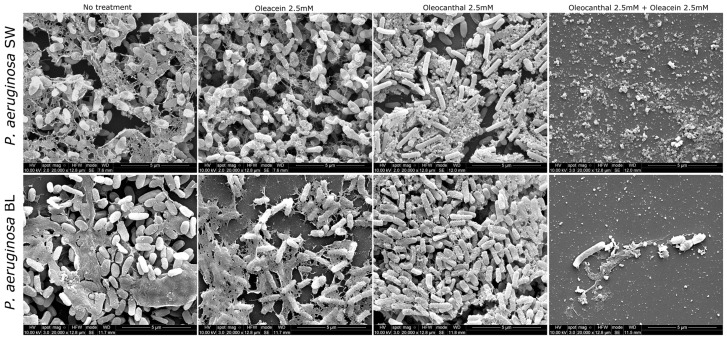
Representative scanning electron micrographs of the anti-biofilm activities of oleacein and oleocanthal against *P. aeruginosa* isolates at the 5 μm magnification level. SW, surgical wound (intermediate drug resistance); BL, bronchoalveolar lavage fluid (multidrug resistance).

**Figure 4 ijms-25-05051-f004:**
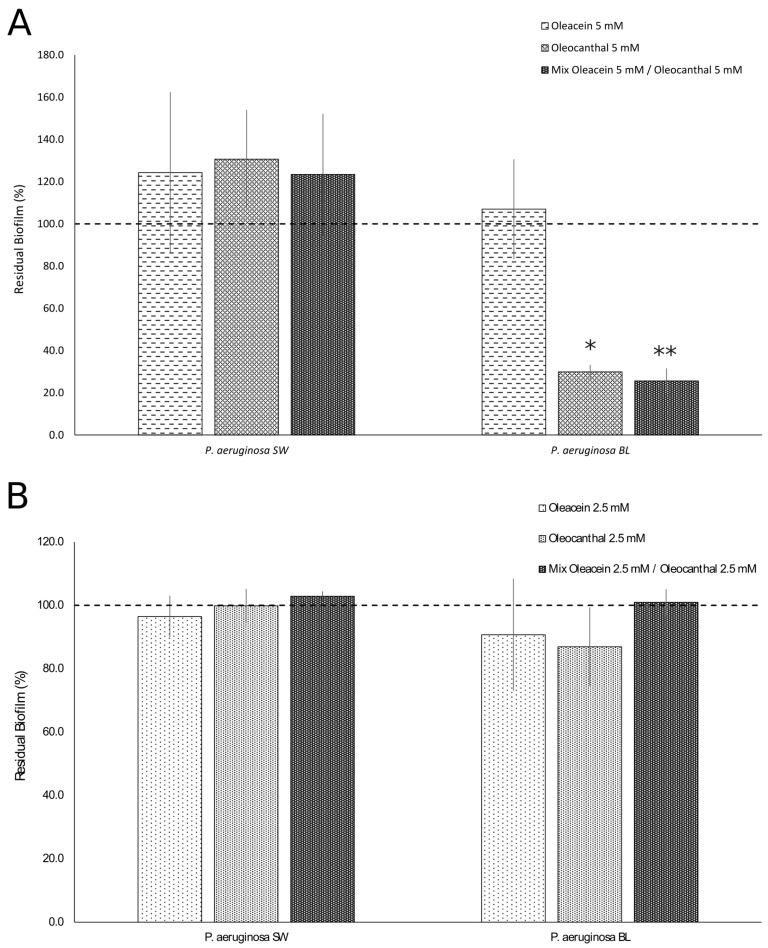
Effects of oleacein and oleocanthal and their combination toward mature biofilm of *P. aeruginosa* isolates. Oleacein, oleocanthal, and their mixture were tested at 5 mM (**A**) and 2.5 mM (**B**) concentrations. SW, surgical wound (intermediate drug resistance); BL, bronchoalveolar lavage fluid (multidrug resistance). Data are expressed as percentages of residual biofilms ± standard deviation as compared to untreated controls. Dotted lines represent the percentage of residual biofilm in untreated *P. aeruginosa*. *, *p* < 0.0001; **, *p* < 0.00001 vs. untreated cells.

**Figure 5 ijms-25-05051-f005:**
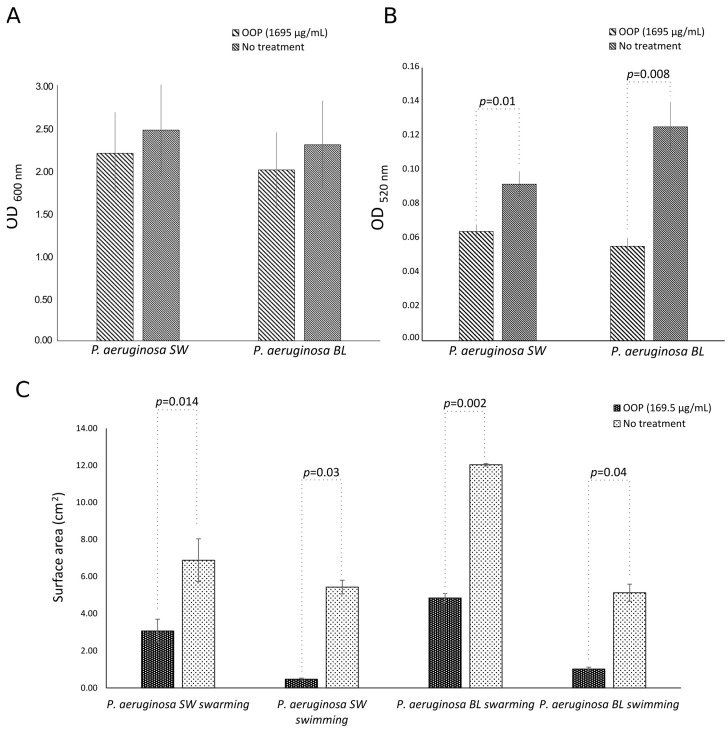
Effects of OOP in NaDES on the productions of alginate (**A**) and pyocyanin (**B**) and on the motilities of *P. aeruginosa* isolates (**C**). SW, surgical wound (intermediate drug resistance); BL, bronchoalveolar lavage fluid (multidrug resistance).

**Figure 6 ijms-25-05051-f006:**
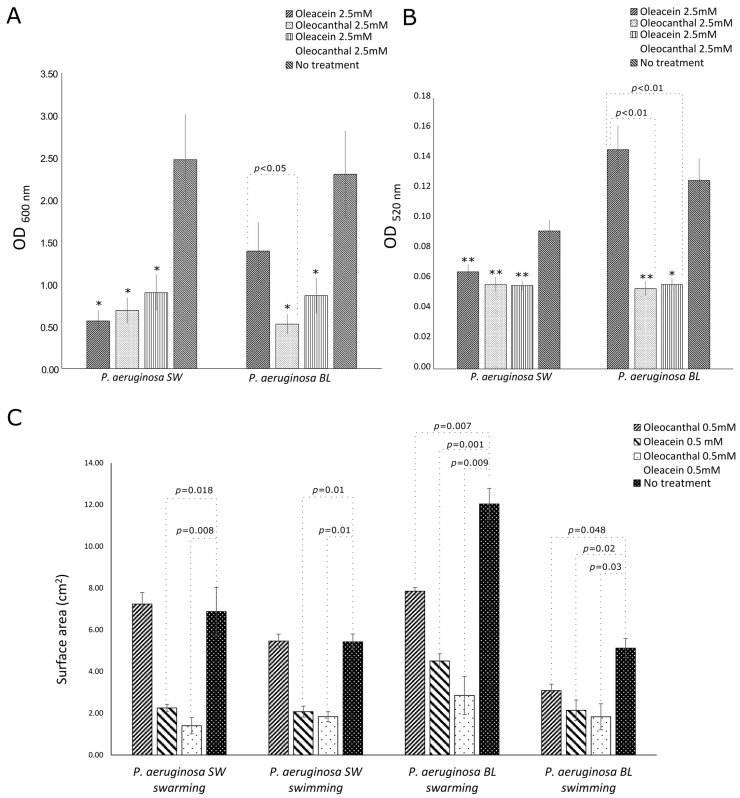
Activities of oleacein, oleocanthal, and their combination in NaDES on the productions of alginate (**A**) and pyocyanin (**B**) and on the motilities of *P. aeruginosa* isolates (**C**). SW, surgical wound (intermediate resistance); BL, bronchoalveolar lavage (multidrug resistance). *, *p* < 0.05 and **, *p* < 0.01 vs. no treatment.

**Table 1 ijms-25-05051-t001:** Phenotypic characterization of clinical isolates of *P. aeruginosa*.

*P. aeruginosa* Strain	Phenotype	Biofilm at 24 h OD 590 nm	Biofilm at 48 h OD 590 nm	Alginate OD 600 nm	Pyocyanin OD 520 nm	Swarming Area (cm^2^)	Swimming Area (cm^2^)
PA BL	Mucoid	1.05 ± 0.59	0.81 ± 0.22	2.16 ± 0.33	0.12 ± 0.01	12.03 ± 0.06	5.11 ± 0.46
PA SW	Non-Mucoid	1.29 ± 0.87	1.10 ± 0.28	1.46 ± 0.17	0.09 ± 0.007	6.87 ± 1.16	5.42 ± 0.37
*p* values		NS	NS	0.048	0.037	0.005	NS

OD, optical density; NS, not significant.

## Data Availability

The raw data supporting the conclusions of this article will be made available by the authors on request.

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
