# Peer review of "Anti-Biofilm Activity of Oleacein and Oleocanthal from Extra-Virgin Olive Oil toward Pseudomonas aeruginosa"

_ijms, 2024, doi:10.3390/ijms25095051_

Round 1

Reviewer 1 Report

Comments and Suggestions for Authors

In the draft of the publication I received,  the convention of using superscript to denote organism numbers to the power of 10 and subscript to denote number of atoms in molecules has not been applied throughout (eg line 105: 1 x  108?, line 184: OsO4?). Whether or not this is simply a computer glitch, it may require attention.

In general the publication is clearly written, interesting and important. The potential for further work might have been usefully added to the Discussion. The following points were noted for attention.

Line 14: This statement is unusual and perhaps needs further explanation or (better) removed.

Lines 119,120: These references are not provided in the References section (?)

Line 213: suggest 'higher' rather than 'high' if a comparison between the two strains is intended

For clarity Figure 1 requires the addition of A and B to match the text.

Figure 2 : the boxes identifying the treatment agents are too small (could they be enlarged slightly, perhaps x4?)

Line 260: suggest 'of' to replace 'by'

Line 281: I presume several scans of each biofilm were examined and those presented in Figure 3 are typical of what was seen

Lines 306, 317: Not clear what 'unlikely' refers to here, suggest delete the word.

Line 367:  a mucoid ... add  'a'

Line 370: In literature, there are evidence... replace 'are' with 'is'

Comments on the Quality of English Language

The use of English was generally good, a few points were noted in the above section. There is a general trend in scientific writing to omit the definite and indefinite articles ('the' and 'a') which does not improve the readability. There was some signs of this in the Discussion section.

Author Response

Reviewer’s report (1)

Authors’ response 

In the draft of the publication I received, the convention of using superscript to denote organism numbers to the power of 10 and subscript to denote number of atoms in molecules has not been applied throughout (eg line 105: 1 x  108?, line 184: OsO4?). Whether or not this is simply a computer glitch, it may require attention.

We are very thankful to the reviewer for pointing out the error. We have reviewed the entire manuscript and corrected all errors related to superscripts and/or subscripts

In general the publication is clearly written, interesting and important. The potential for further work might have been usefully added to the Discussion. The following points were noted for attention.

Following the reviewer's suggestion, we have added a passage at the end of the Discussion section, expressing the need for future studies to investigate the cellular and molecular mechanisms through which oleacein and oleocanthal exert the effects described in this manuscript.

Line 14: This statement is unusual and perhaps needs further explanation or (better) removed.

This statement is accepted by IJMS when the first and second authors equally contribute to the experimental and writing part of the study, as already present in other papers from this Journal (Tang, J et al. Int. J. Mol. Sci. 2024, https://doi.org/10.3390/ijms25084493).

Lines 119,120: These references are not provided in the References section (?)

We are sorry for the mistake, and we revised the reference numbers and list.

Line 213: suggest 'higher' rather than 'high' if a comparison between the two strains is intended

We thank the Reviewer for bringing the error to our attention. We have made the necessary corrections to the text.

For clarity Figure 1 requires the addition of A and B to match the text.

We are sorry for the mistake, and we revised the text to match Figure 1 that does not have two panels.

Figure 2 : the boxes identifying the treatment agents are too small (could they be enlarged slightly, perhaps x4?)

We enlarged the boxes identifying the treatment agents in Figure 2, as suggested by the Reviewer.

Line 260: suggest 'of' to replace 'by'

We replaced “by” with “of”.

Line 281: I presume several scans of each biofilm were examined and those presented in Figure 3 are typical of what was seen

We have added this information in the legend of Figure 3.

Lines 306, 317: Not clear what 'unlikely' refers to here, suggest delete the word.

Thanks a lot for the suggestion. We removed the words.

Line 367:  a mucoid ... add  'a'

Thanks a lot for the suggestion. We performed the correction.

Line 370: In literature, there are evidence... replace 'are' with 'is'

Thanks a lot for the suggestion. We performed the correction.

Reviewer 2 Report

Comments and Suggestions for Authors

Dear editor

·        The MS’ Anti-biofilm activity of oleacein and oleocanthal from Extra  Virgin Olive Oil towards Pseudomonas aeruginosa”, The MS represents some novel premiminary data concerning the antibiofilm and antivirulence activity of oleacein and oleocanthal, however, some main points need to be added and clarified. Mainly. All the data are phenotypic study there is no genotypic study on the genetic level such as study the effect on gene expression By RT-PCR.

·        Also, time kill study need to be added to confirm that the used oleacein and oleocanthal  at sub-MIC inhibit virulence factors NOT killed bacteria.

·        The authors need to study the effect on global regulators of virulence factors such as Qs in P. aeruginosa

Also, plagiarism of the MS need to be revised as the percentage similarity in some parts is high. 

Comments

·        Line 109; what do you mean polyphenol concentrations from 6780 μg/ml to 424 μg/ml

·        Line 119; Ref (Krish- 119 namoorthy et al., 2018; Wijesundara et al. 2021) were Not indicated as numbers

·        In the biofilm assay; what was the conc of oleacein and oleocanthal use, is it sub MIC or what X MIC? Please specify in the method.

·        In the biofilm assay methods, how the planktonic cells were fixed to the wells before staining ?

·        In the biofilm assay methods , Add the suitable ref of the method

·        The Phenotypic characterization of clinical isolates of P. aeruginosa is mentioned in the result section NOT in the methods, please add.

·        How can you explain that the biofilm of the mucoid isoltes after 48 g is less than that after 24 hr, table 1?

·        Add the detailed data of antibiotic susceptibility patterns of the tested isolates. Name, conc of the used antibiotics and R OR S.

·        What is the bases of the classification of the tested isolates to multidrug resistant (PA BL) and intermediately resistant (PA SW).

·        Specify the conc of oleacein and oleocanthal used in Scanning electron microscopy

·        Why stronger inhibitory effect was observed against PA BL (71.4%) as compared to PA SW (33%), explain.

·        In the figure 2, Why the author specify that data did not reach statistical significance, however, P<0.001 which is highly significant?

·        Data organization, the arrangement of the method section should be similar to that of the result section.

·        Figure 2, how the residual biofilm percentage in the panel A differs from that of B although the same isolates.

·        Figure 3, need to add the solvent figure to eliminate out its effect.

·        Figure 4, the sub MICs of oleacein and oleocanthal did not affect mature biofilm why the authors did not try to use higher conc.

·        Figure 5 and 6 need to be combined for the same test represent all concentrations of the used compounds or its combinations.

·        All the data are phenotypic study there is no genotypic study on the genetic level such as study the effect on gene expression By RT-PCR.

·        Also, time kill study need to be added to confirm that the used oleacein and oleocanthal  at sub-MIC inhibit virulence factors NOT killed bacteria.

·        The authors need to study the effect on global regulators of virulence factors such as Qs in P. aeruginosa

·        The discussion section did not provide explanation of the result.

Comments on the Quality of English Language

Dear editor

·        The MS’ Anti-biofilm activity of oleacein and oleocanthal from Extra  Virgin Olive Oil towards Pseudomonas aeruginosa”, The MS represents some novel premiminary data concerning the antibiofilm and antivirulence activity of oleacein and oleocanthal, however, some main points need to be added and clarified. Mainly. All the data are phenotypic study there is no genotypic study on the genetic level such as study the effect on gene expression By RT-PCR.

·        Also, time kill study need to be added to confirm that the used oleacein and oleocanthal  at sub-MIC inhibit virulence factors NOT killed bacteria.

·        The authors need to study the effect on global regulators of virulence factors such as Qs in P. aeruginosa

Comments

·        Line 109; what do you mean polyphenol concentrations from 6780 μg/ml to 424 μg/ml

·        Line 119; Ref (Krish- 119 namoorthy et al., 2018; Wijesundara et al. 2021) were Not indicated as numbers

·        In the biofilm assay; what was the conc of oleacein and oleocanthal use, is it sub MIC or what X MIC? Please specify in the method.

·        In the biofilm assay methods, how the planktonic cells were fixed to the wells before staining ?

·        In the biofilm assay methods , Add the suitable ref of the method

·        The Phenotypic characterization of clinical isolates of P. aeruginosa is mentioned in the result section NOT in the methods, please add.

·        How can you explain that the biofilm of the mucoid isoltes after 48 g is less than that after 24 hr, table 1?

·        Add the detailed data of antibiotic susceptibility patterns of the tested isolates. Name, conc of the used antibiotics and R OR S.

·        What is the bases of the classification of the tested isolates to multidrug resistant (PA BL) and intermediately resistant (PA SW).

·        Specify the conc of oleacein and oleocanthal used in Scanning electron microscopy

·        Why stronger inhibitory effect was observed against PA BL (71.4%) as compared to PA SW (33%), explain.

·        In the figure 2, Why the author specify that data did not reach statistical significance, however, P<0.001 which is highly significant?

·        Data organization, the arrangement of the method section should be similar to that of the result section.

·        Figure 2, how the residual biofilm percentage in the panel A differs from that of B although the same isolates.

·        Figure 3, need to add the solvent figure to eliminate out its effect.

·        Figure 4, the sub MICs of oleacein and oleocanthal did not affect mature biofilm why the authors did not try to use higher conc.

·        Figure 5 and 6 need to be combined for the same test represent all concentrations of the used compounds or its combinations.

·        All the data are phenotypic study there is no genotypic study on the genetic level such as study the effect on gene expression By RT-PCR.

·        Also, time kill study need to be added to confirm that the used oleacein and oleocanthal  at sub-MIC inhibit virulence factors NOT killed bacteria.

·        The authors need to study the effect on global regulators of virulence factors such as Qs in P. aeruginosa

·        The discussion section did not provide explanation of the result.

Author Response

Reviewer’s comments (2)

Authors’ answers

Also, time kill study need to be added to confirm that the used oleacein and oleocanthal at sub-MIC inhibit virulence factors NOT killed bacteria.

We thank the Reviewer for the suggestion, and we added data about the time kill of P. aeruginosa strains at sub-MIC concentrations of oleacein and oleocanthal (see page 4, lines 160-168, page 6, lines 254-258, and supplementary Figure S1).

The authors need to study the effect on global regulators of virulence factors such as Qs in P. aeruginosa

We thank the Reviewer for the suggestion, and our next step will be to carry out more research on this aspect.

In the actual scenario of increasing multi-drug resistance amongst bacterial infections, our aim was to first discover novel natural molecules, in particular oleacein and oleocanthal from Extra Virgin Olive Oil, with anti-biofilm activity towards clinical isolates of P. aeruginosa with varying degree of antibiotic resistance. In particular, P. aeruginosa is an ESKAPE pathogen with the most prevalence in hospital environments, against which new antibiotics are urgently needed. We also investigated some virulence factors regulated by the QS of P. aeruginosa and involved in biofilm formation, including pyocyanin and motility (Khayat MT et al. doi: 10.3390/biomedicines11051442; Papa R et al. doi: 10.3390/antibiotics10080944), from a phenotypic viewpoint.

Also, plagiarism of the MS need to be revised as the percentage similarity in some parts is high.

We revised the entire manuscript in order to solve the plagiarism issues.

Line 109; what do you mean polyphenol concentrations from 6780 μg/ml to 424 μg/ml

The anti-bacterial activity of OPP (total polyphenol content of the stock solution 16950 μg/ml, see page 2, lines 77-78) was assayed at polyphenol concentrations starting from 6780 μg/ml up to 424 μg/ml, following 2-fold serial dilutions (6780 – 3390 – 1695 – 847.5 – 423.75 μg/ml). We modified Materials and Methods section, paragraph 2.4, page 4, lines 148-150.

Line 119; Ref (Krish- 119 namoorthy et al., 2018; Wijesundara et al. 2021) were Not indicated as numbers

We are sorry for the mistake, and we revised the reference numbers and list.

In the biofilm assay; what was the conc of oleacein and oleocanthal use, is it sub MIC or what X MIC? Please specify in the method.

The concentrations of oleacein and oleocanthal used in the biofilm assay were based on MIC results and were chosen as “sub-MIC”. This information is reported in Results section, paragraph 3.3, page 6, lines 269-270.

In the biofilm assay methods, how the planktonic cells were fixed to the wells before staining?

Planktonic cells are removed during the washing steps of the protocol for biofilm staining, as also reported by Papa R et al., Int J Mol Sci 2020, doi: 10.3390/ijms21239258.

In the biofilm assay methods, Add the suitable ref of the method

We are sorry for the mistake, and we added the relevant reference “Papa R et al., 2020, doi: 10.3390/ijms21239258” (see page 3, line 115).

The Phenotypic characterization of clinical isolates of P. aeruginosa is mentioned in the result section NOT in the methods, please add.

We have added the relevant paragraphs in Materials and Methods section, describing the phenotypic characterization of P. aeruginosa isolates (see pages 2-4, lines 94-142).

How can you explain that the biofilm of the mucoid isoltes after 48 g is less than that after 24 hr, table 1?

The differences between the 48h biofilm against the 24h biofilm, reported in Table 1, are not statistically significant.

The slight variability in biofilm amount from 24 h to 48 h may be related to dynamic changes in the biofilm structure during its lifecycle (Rubio-Canalejas A et al. doi: 10.3389/fmicb.2022.959156). Also, variations in the culture surfaces amongst different wells in the same microplate or in different microplates might account for the differences.

Add the detailed data of antibiotic susceptibility patterns of the tested isolates. Name, conc of the used antibiotics and R OR S.

We thank the Reviewer for the suggestion, and we have added the table with the detailed data on the antibiotic susceptibility patterns of tested isolates (see Supplementary table S1).

What is the bases of the classification of the tested isolates to multidrug resistant (PA BL) and intermediately resistant (PA SW).

As reported in Paragraph 2.3.1. “Antibiotic Susceptibility Assay”, the P. aeruginosa strain BL was defined as multi-drug resistance, in accordance to European Centre for Disease Prevention and Control [Magiorakos AP et al. doi:10.1111/j.1469-0691.2011.03570.x], since it showed resistance, overall, to nine antibiotics belonging to five categories. The P. aeruginosa strain SW was defined as intermediately resistant since it showed an intermediate susceptibility to approximately 50% of all antibiotics tested (see page 6, lines 233-235, and Supplementary Table S1).

Specify the conc of oleacein and oleocanthal used in Scanning electron microscopy

The concentrations of oleacein and oleocanthal used in Scanning electron microscopy analysis are reported in Results section, page 7, lines 288-297, and in Figure 3.

Why stronger inhibitory effect was observed against PA BL (71.4%) as compared to PA SW (33%), explain.

The different effect of oleacein against the biofilm formation observed in PA BL and PA SW may be linked to differences amongst the bacterial strains in their genetic background and in the pro-duction of other polysaccharides (polysaccharide synthesis locus) and molecules (pyoverdine and rhamnolipids), as well as in the presence of extracellular DNA into the bio-film matrix. All of these factors are involved mainly in the initial attachment of bacterial cells and early biofilm formation (Campoccia et al., 2021; Bonincontro et al., 2023).

In the figure 2, Why the author specify that data did not reach statistical significance, however, P<0.001 which is highly significant?

In Figure 2, we reported statistical significance only when comparing the effect of the tested compounds against untreated bacterial cells (as indicated by the asterisks in the figure 2 legend). All the other comparisons (e.g. between different compounds and strains) did not reach statistical significance, as also reported in Results section, paragraph 3.3, page 6, lines 269-287.

Data organization, the arrangement of the method section should be similar to that of the result section.

We thank the Reviewer for the suggestion, and we revised the Materials and Methods for arranging this section similarly to the Results section.

Figure 2, how the residual biofilm percentage in the panel A differs from that of B although the same isolates.

We are sorry for the misunderstanding. We modified figure 2 including A and B, and the legend, highlighting the different concentrations used. In fact, the difference in residual biofilm percentage between panel A and B results from the different concentrations of oleacein, oleocanthal and their mix used in the experiments.

Figure 3, need to add the solvent figure to eliminate out its effect.

In our preliminary experiments, we demonstrated that the solvent NaDES had no anti-biofilm activity against both strains. Hence, it was not assayed for the scanning electron microscopy experiment. We added the relevant statement in Results section, paragraph 3.3 (see page 6, lines 260-262).

Figure 4, the sub MICs of oleacein and oleocanthal did not affect mature biofilm why the authors did not try to use higher conc.

The highest concentration used in our experiments was MIC/2 (5 mM); at this concentration, oleocanthal and the mix oleacein/oleocanthal were both effective exclusively against the mature biofilm of PA BL strain. Differently, oleacein showed no effect towards mature biofilm in both P. aeruginosa strains.

We tested high concentrations in preliminary experiments, up to 25 mM (2.5X MIC), evidencing a similar activity profile as above described against mature biofilm.

Figure 5 and 6 need to be combined for the same test represent all concentrations of the used compounds or its combinations.

Figure 5 represents the effects of the extra virgin olive oil extract (OOP), whereas Figure 6 represents the effects of purified oleacein and oleocanthal, toward PA BL and PA SW strains. Given the considerable number of compounds and conditions tested, combining the two figures together could make it more complex to read and interpret the results. Also, we received the request, by the other Reviewer, to enlarge the size of the boxes in the Figures.

All the data are phenotypic study there is no genotypic study on the genetic level such as study the effect on gene expression By RT-PCR.

As above described, our aim was first to assess that our natural molecules possessed anti-biofilm activity towards clinical isolates of P. aeruginosa. Our next step will be to dig deeper into the mechanisms underlying the antibiofilm and antibacterial activities of oleacein and oleocanthal by performing genotypic and gene expression studies, as suggested by the Reviewer.

Also, time kill study need to be added to confirm that the used oleacein and oleocanthal at sub-MIC inhibit virulence factors NOT killed bacteria.

We thank the Reviewer for the suggestion, and we added data about the time kill of P. aeruginosa strains at sub-MIC concentrations of oleacein and oleocanthal (see supplementary Figure S1).

The authors need to study the effect on global regulators of virulence factors such as Qs in P. aeruginosa

We thank the Reviewer for the suggestion, and our next step will be to carry out more research on this aspect.

In the actual scenario of increasing multi-drug resistance amongst bacterial infections, our aim was to first discover novel natural molecules, in particular oleacein and oleocanthal from Extra Virgin Olive Oil, with anti-biofilm activity towards clinical isolates of P. aeruginosa with varying degree of antibiotic resistance. In particular, P. aeruginosa is an ESKAPE pathogen with the most prevalence in hospital environments, against which new antibiotics are urgently needed. We also investigated some virulence factors regulated by the QS of P. aeruginosa and involved in biofilm formation, including pyocyanin and motility (Khayat MT et al. doi: 10.3390/biomedicines11051442; Papa R et al. doi: 10.3390/antibiotics10080944), from a phenotypic viewpoint.

The discussion section did not provide explanation of the result.

We thank the Reviewer, and we expanded the discussion about our results including the recommendation of the Reviewer (see page 14, lines 389-398).

Round 2

Reviewer 2 Report

Comments and Suggestions for Authors

The authors performed the required corrections

More data discussion need to be added

Comments on the Quality of English Language

The authors performed the required corrections

More data discussion need to be added